

# The effect of changing topography on the coordinated marching of locust nymphs

Guy Amichay[1,4], Gil Ariel[2] and Amir Ayali[1,3]

[1] Department of Zoology, Tel Aviv University, Tel Aviv, Israel
[2] Department of Mathematics, Bar-Ilan University, Ramat Gan, Israel
[3] Sagol School of Neuroscience, Tel Aviv University, Tel Aviv, Israel
[4] Current affiliation: The Department of Collective Behaviour at the University of Konstanz/Max Planck Institute of Ornithology, Konstanz, Germany

## ABSTRACT

Collective motion has traditionally been studied in the lab in homogeneous, obstacle-free environments, with little work having been conducted with changing landscapes or topography. Here, the impact of spatial heterogeneity on the collective motion exhibited by marching desert locust nymphs was studied under controlled lab conditions. Our experimental circular arenas, incorporating a funnel-like narrowing followed by re-widening, did not constitute a major barrier to the locusts but, rather, mimicked a changing topography in the natural environment. We examined its effects on macroscopic features of the locust collective behavior, as well as the any changes in their marching kinematics. A major finding was that of the limited extent to which the changing topography affected system-level features of the marching locust group, such as the order parameter and the fraction of walking individuals, despite increased crowding at the funnel. Overall, marching kinematics was also very little affected, suggesting that locust marching bands adjust to the environment, with little effect on the overall dynamics of the group. These findings are in contrast to recent theoretical results predicting that environmental heterogeneities qualitatively alter the dynamics of collectively moving particles; and highlight the crucial role of rapid individual plasticity and adaptability in the dynamics of flocks and swarms. Our study has revealed other important features of the marching behavior of the desert locust in addition to its robustness: the locusts demonstrated both, clear thigmotaxis and a tendency to spread-out and fill the available space.

Subjects Animal Behavior, Biogeography, Ecology, Entomology
Keywords Collective motion, Swarming, Environmental effects, Topological changes, Spatial heterogeneity, *Schistocerca gregaria*, Locust marching bands

## INTRODUCTION

Locust swarming constitutes a unique model of animal collective motion (recent review in *Ariel & Ayali, 2015*). Collective motion, being a dominant feature of the behavior of these insects, is also responsible for their notorious reputation as a major pest. Of particular importance is the behavior of the swarms of locust nymphs, comprising millions of insects forming marching bands that may cover vast areas of land, stretching across hills and valleys as far as the eye can see (e.g., *Ellis & Ashall, 1957*; *Uvarov, 1977*; *Hunter, McCulloch & Spurgin, 2008*; our own personal observations of the desert locust in Israel's Negev desert, spring 2013).

Corresponding author
Amir Ayali, ayali@post.tau.ac.il

Different aspects of locust motion, the interactions between conspecifics and their impact on the build-up and maintenance of coordinated movement, have been studied using controlled laboratory experiments (e.g., *Ellis, 1951*; *Buhl et al., 2006*; *Bazazi et al., 2012*; *Ariel & Ayali, 2015*, and references therein). In recent years, advanced tracking algorithms have been utilized to monitor up to several dozen animals at a time at high spatio-temporal resolution for later off-line analysis (e.g., *Buhl et al., 2006*; *Ariel et al., 2014*). However, those studies were limited to homogeneous, obstacle-free environments, in order to facilitate emergence of the locust collective motion. Such an environment clearly differs from that encountered by the marching locust swarm (or by any other animals demonstrating mass terrestrial movement) in natural conditions, which may feature a complex terrain, vegetation etc., obscuring and obstructing the path of the locusts.

The effects of an heterogeneous environment on the ability of large systems of moving animals (or particles) to form collective dynamic patterns have been mostly studied in simulations. Theoretical work, using simplified models of collective motion, has predicted that the effect of spatial heterogeneities on the ability of swarms to organize and form synchronized motion is highly non-trivial and to some extent non-intuitive. For example, obstacles may significantly change the phase diagram of the system, either disrupting or promoting coherent motion, depending on obstacle density, noise intensity, and other model parameters (*Hatzikirou & Deutsch, 2008*; *Chepizhko et al., 2013*; *Chepizhko & Peruani, 2015*; *Quint & Gopinathan, 2015*). Boundaries can promote the formation of vortices (*Duparcmeur, Herrmann & Troadec, 1995*; *Czirok et al., 1996*; *Szabo et al., 2006*; *Barbaro et al., 2009*; *Potiguar, Farias & Ferreira, 2014*), induce traffic-lane patterns (*Hernandez-Ortiz, Underhill & Graham, 2009*; *Ariel et al., 2013*) and introduce heterogeneity in densities (*Drocco, Reichhardt & Reichhardt, 2012*; *Hernandez-Ortiz, Stoltz & Graham, 2005*). Several theoretical studies have considered the effect of heterogeneous environments on specific animal systems, such as birds (analyzing risks of collision with wind turbines; *Croft et al., 2015*), fish (predicting the location of Capelin off the Icelandic shores; *Barbaro et al., 2009*), and living cells (studying the collective cell dynamics of tissue cells in micro-fabricated arenas; *Szabo et al., 2006*). The study of pedestrian flow through bottlenecks has a rich history of over 100 years (e.g., *Helbing, 2001*; *Appert-Rolland et al., 2009*). Some of the basic phenomena reported to be associated with a locally reduced mobility or increased density include a drop in flow capacity (depending on bottleneck width), division into free-flowing and congested areas, appearance of shock-like waves, and more. Ample research has also been conducted on the interactions of swarms of autonomous agents (robots) with obstacles (e.g., *Trianni, Nolfi & Dorigo, 2006*) and of virtual particles, within the scope of swarm optimization methods (e.g., *Shklarsh et al., 2011*).

In contrast to the theoretical studies discussed above, relatively few experimental reports are available on the macroscopic dynamics of collectively moving organisms within an heterogeneous environment. These include microbial swarms interacting within complex substrates (*Tuson & Weibel, 2013*), tissue cells in micro-fabricated arenas (motivated by the highly heterogeneous media that form the extracellular matrix; e.g., *Friedl & Broecker, 2004*; *Szabo et al., 2006*), and pedestrians at bottlenecks, demonstrating a continuous reduction
in flow capacity following a reduction in bottleneck width (*Hoogendoorn & Daamen, 2005*; *Appert-Rolland et al., 2009*). Experiments using different species of ants have shown that topology can have varying effects on the collective movement of the different species. For example, it has been shown that in *Atta cephalotes* (*Burd et al., 2002*), increased density due to a reduction in trail width reduces the mean traveling speed and the flux of ants across a barrier. In contrast, the flow of black garden ants (*Lasius niger*) does not seem to depend on trail width (*Dussutour et al., 2004*). To the best of our knowledge, locusts, as well as other animal models of collective motion, have been little studied in this respect. The movement of ants along trails fundamentally differs from that of locusts, as ants seldom turn while walking, being typically either non laden when exiting the nest, or laden when returning to it (*Burd et al., 2002*; *Dussutour et al., 2004*). In contrast, the frequency of turning (when changing from a standing to a walking state) has been found to be central in the ability of locusts to form and maintain collective movement (*Ariel et al., 2014*).

This current study sought to investigate locust marching behavior in a changing topography: in our case, a circular arena incorporating a funnel-like narrowing followed by a re-widening. This spatial heterogeneity did not constitute a major barrier to the locusts but, rather, introduced certain simple features that are also found in natural environments, such as a path among large rocks or simply a narrowing gorge. We were interested in examining the effects of such topographical changes on macroscopic features of the locust collective behavior, such as the order parameter, as well as any resultant changes in statistics of the marching behavior of individuals (e.g., speed, walking-bout duration etc.), and in the dynamics of local interactions between the insects, such as distance between neighboring marching locusts, and how these, in turn, translate to swarm dynamics.

One of the key questions regarding the response of the individual to a changing environment is that of the effect of topography on the coordinated movement of the crowd. Specifically, it is not clear whether such a response is universal, i.e., common to all swarms, from single cells to locusts and humans; or, alternatively, it significantly differs across species. The main goal of the current study was to revisit previous experimental and theoretical work and to provide novel insights into the effect of a specific type of heterogeneous environment on the collective dynamics of marching locusts. In particular, we sought to unravel the mechanisms by which individual locusts alter their kinematic parameters in order to adjust to the changing environment and diminish its effect on the swarm.

## MATERIALS AND METHODS

### Animals

Desert locusts, *Schistocerca gregaria* (Forskål), were obtained from our colony at the Department of Zoology, Tel Aviv University, Israel. The locusts, approaching the gregarious phase, were reared over many generations in 60-liter metal cages at a density of 100–160 animals per cage, under a controlled temperature of 30 °C and 35–60% humidity, and a 12D:12L cycle. Additional radiant heat was provided by 25W incandescent electric bulbs during daytime to reach a final day temperature of 35–37 °C. The locusts were fed daily
with wheat seedlings and dry oats. All experiments were performed on nymphs of the final (Vth) nymphal-instar (3–4 cm in length and ∼0.5 cm in width).

## Behavioral setup

We conducted a series of experiments in each of which a group of nymphs was allowed to move freely in a circular arena. The basic experimental arena was composed of a flat blue Perspex sheet circumscribed by an outer flexible blue plastic wall (60 cm diameter × 55 cm high). An inner circular wall made of similar plastic (diameter 30 cm) was inserted to create a ring shaped sphere (Fig. 1A). The different experimental conditions were achieved by either leaving the inner circle in the middle (no displacement), or moving it by 5 or 10 cm sideways towards the outer wall (creating a funnel 10 or 5 cm wide, respectively; see Fig. 1B). The lower 10 cm of the arena walls were thinly coated with Fluon (Whitford Plastics Ltd, Runcorn, UK) to prevent the nymphs from climbing. The arena was placed in our temperature-controlled room (30 °C) and lit from above by a 100 W ring bulb. The propensity of locusts to swarm in an experimental arena is highly sensitive to the number of animals introduced into it (Buhl et al., 2006). Hence, fifty nymphs were introduced into the arena in each experiment, which is a sufficiently high density to facilitate the formation of synchronized movement (Ellis, 1951; Buhl et al., 2006; Buhl et al., 2011; Ariel et al., 2014), and also approximates the range of densities observed in the field (A Ayali et al., 2013, unpublished data). The locusts were continuously monitored and their movement recorded, using a Sony HDR-XR550E digital camera with a 30 fps rate, for later off-line analysis. Using a custom-designed continuous multiple-target tracking method (Ariel et al., 2014, Fig. 1C), we simultaneously tracked the movement of all individuals at high spatio-temporal resolution (full HD allowed a detailed analysis of the behavior of each individual). We conducted overall 15 experiments, 5 under each experimental condition (inner circle displacement of 0, 5, or 10 cm). Each experiment lasted approximately 4 h, from which the last 200,000 frames were selected (ca. 2 h), for consistency between experiments and to ensure that we had captured robust locust marching. These were then divided into one hour blocks, providing ample data for statistics. A 10 min period (which is several times longer than the typical resting or walking duration) at the end of each block was discarded to ensure that the blocks were independent of one another. Division into a smaller number of blocks did not qualitatively change the results. Hence, we were left with two sections from each of the five experiments, under each of three experimental condition, totaling 30 experimental sections for further comparative analysis ($n = 10$ for each experimental condition).

## Analysis of behavior

All data analysis was performed in the MATLAB environment (MathWorks, Natick, Massachusetts, USA). To minimize bias wherever possible, blinded methods were used to analyze data. Following smoothing of trajectories as detailed in Ariel et al. (2014), specific attributes of the system and the individual locusts were defined and analyzed as follows.
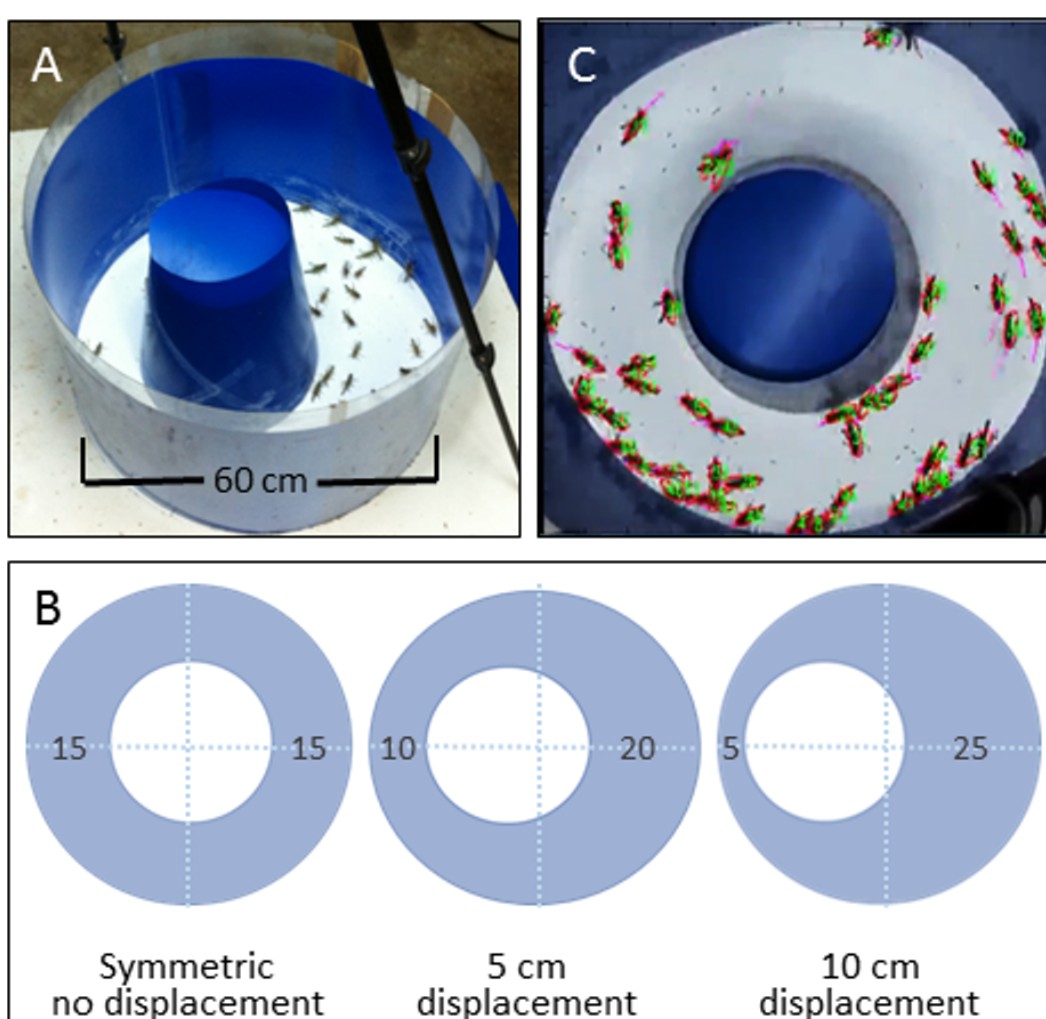

**Figure 1 The behavioral setup.** (A) A top view of the experimental circular arena. (B) The different experimental settings: displacement of the inner circle results in different-width funnel-like structures. The width of the different sections of the arena are noted. (C) A snap-shot from a video monitoring of the locusts' collective motion. All nymphs are assigned numbers and their velocities are noted.

### Identification of walking bouts

An animal was classified as *walking* if it was moving at a speed greater than 2 mm/sec for at least 11 frames. We denote by $w_k(t) = 0$, 1 or $-1$ if at time $t$ animal $k$ is standing, walking in the counter clockwise (CCW) direction, or walking in the clockwise (CW) direction, respectively. We denote by $f(t)$ the *fraction of moving animals* at time $t$,

$$f(t) = \frac{1}{N} \sum_k |w_k(t)|,$$

where $N$ is the number of animals in the arena. Thus, $0 \leq f(t) \leq 1$, where $f(t) = 0$ implies that all animals are standing, while $f(t) = 1$ implies that they are all walking.

For a given animal, the time series $w_k(t)$ can be divided into periods of walking and resting, i.e., continuous segments in which either $w_k(t) = 1$ or $-1$. Accordingly, we consider the distribution of *walking and resting durations*, denoted $t_{\text{walk}}$ and $t_{\text{rest}}$, respectively.

### Walking speed
The speed of animal $k$ at time $t$ is denoted $v_k(t)$.

### The order parameter
To quantify order and synchronization in the system, we defined the instantaneous order parameter as the average direction (CW or CCW) of walking animals at time $t$. More precisely,

$$\phi(t) = \frac{1}{Nf(t)} \sum_k w_k(t). \tag{1}$$

When no animals are moving, the system does not have a preferred direction and $\phi(t) = 0$. Thus, $|\phi|$ is a measure of the level of coordination within the swarm; a value of $|\phi|$ close to zero indicates that the swarm is disorganized, with animals moving in random directions; while a value close to one indicates a highly synchronized state with all directions aligned. The sign of the order parameter indicates the direction of the swarm: positive $\phi$ refers to CCW motion, while negative $\phi$ indicates CW motion. Note that $\phi$ here differs from the order parameter used by many authors, since it depends only on walking animals.

### Number of neighbors
For each frame, we collected all the locations of all locusts and calculated the number of conspecifics up to a distance of 6 cm from each focal animal (corresponding to circa two locust-body-lengths). The number of neighbors of animal $k$ at time $t$ is denoted $n_k(t)$.

### Number of close moving neighbors
Similar to the number of neighbors, ignoring standing animals. In other words, the number of moving neighbors, $n_{\text{move},k}(t)$, defined only if animal $k$ is moving at time $t$, is the number of conspecifics walking within a distance of 6 cm from animal $k$. Both the number of neighbors and the number of moving neighbors describe the visual stimulus of an animal and have been previously shown to affect the propensity of animals to start moving (*Ariel et al., 2014*).

### Significance tests
In the comparison of the different arena types, the *p*-values (in parenthesis) report the probability that the means of all three arenas are the same (an ANOVA analysis calculated using Matlab's One-way analysis of variance function). In the comparison of the arena halves, the *p*-values (in parenthesis) report the probability that the means obtained from the two halves are the same.

### Correlation coefficients
The correlation coefficient, $\rho$, where noted (between any pair of the measurements described above), is the Pearson coefficient using all data points. Correlations were

considered as significant for $|\rho| > 0.3$, which, for $n = 30$ corresponds to a (one-sided) $P$-value of 0.05 or lower.

### Heat map representations

An image of the arena was divided into 248*248 pixels. For each pixel, we summed all the times that the center of a locust appeared in it (within one experiment). In order to account for differences between the experiments in terms of overall density, we summed the values from all the pixels together and divided the value of each pixel by that sum. In other words, each pixel now had a value representing the fraction of the animals in that experiment. We visualized this by representing each pixel, according to its value, with a specific color.

Because trajectories are smoothed, they are not as accurate as the raw resolution allows (the error is 3–5 pixels, corresponding to about 1 mm). Hence, the reduced resolution used for heat-maps (and only for heat maps) allows convenient representation without any significant loss of accuracy.

### Heat map of movement

Similar to the heat map, with only moving animals taken into account.

## RESULTS

### Changing topography does not affect the global parameters of the locust collective motion

The order parameter, $\phi(t)$, is an accepted measure of synchronization in a system (*Buhl et al., 2006*; *Ariel & Ayali, 2015*). Fig. 2A presents two examples of the evolution of the order parameter and the overall fraction of walking locusts, $f(t)$, during a 50 min time section in our experiments. In accordance with these examples, when plotting the average of $|\phi|$ against $f$ in all experiments, they were consistently found to be highly correlated (Fig. 2B), confirming previous findings regarding the increased tendency of animals to join their conspecifics as the number of walkers increases (*Ariel et al., 2014*). Neither this correlation nor the actual magnitude of $\varphi$ and $f$ were affected by the topography of the arena, i.e., they were independent of the displacement of the inner circle, $d$ (Figs. 2C and 2D, $p$-values 0.57 and 0.53, respectively).

Other attributes of the overall kinematics of the marching locusts were also not affected by the arena topography (Figs. 3). These include the average speed of all locusts in the arena (Fig. 3A, $p$-value 0.99), and the duration of a locust's walking or pausing bout (Figs. 3C and 2D $p$-values 0.4 and 0.41). This is not surprising, as many of the marching kinematics showed a significant correlation with $\varphi$ and with $f$, again irrespective of the arena topography (see for example a plot of speed vs. $f$, Fig. 3B). No differences in density-related parameters, such as the number of locusts or of marching locusts only, within a short range from a focal locust were found (Figs. 3E and 3F $p$-values 0.55 and 0.19).

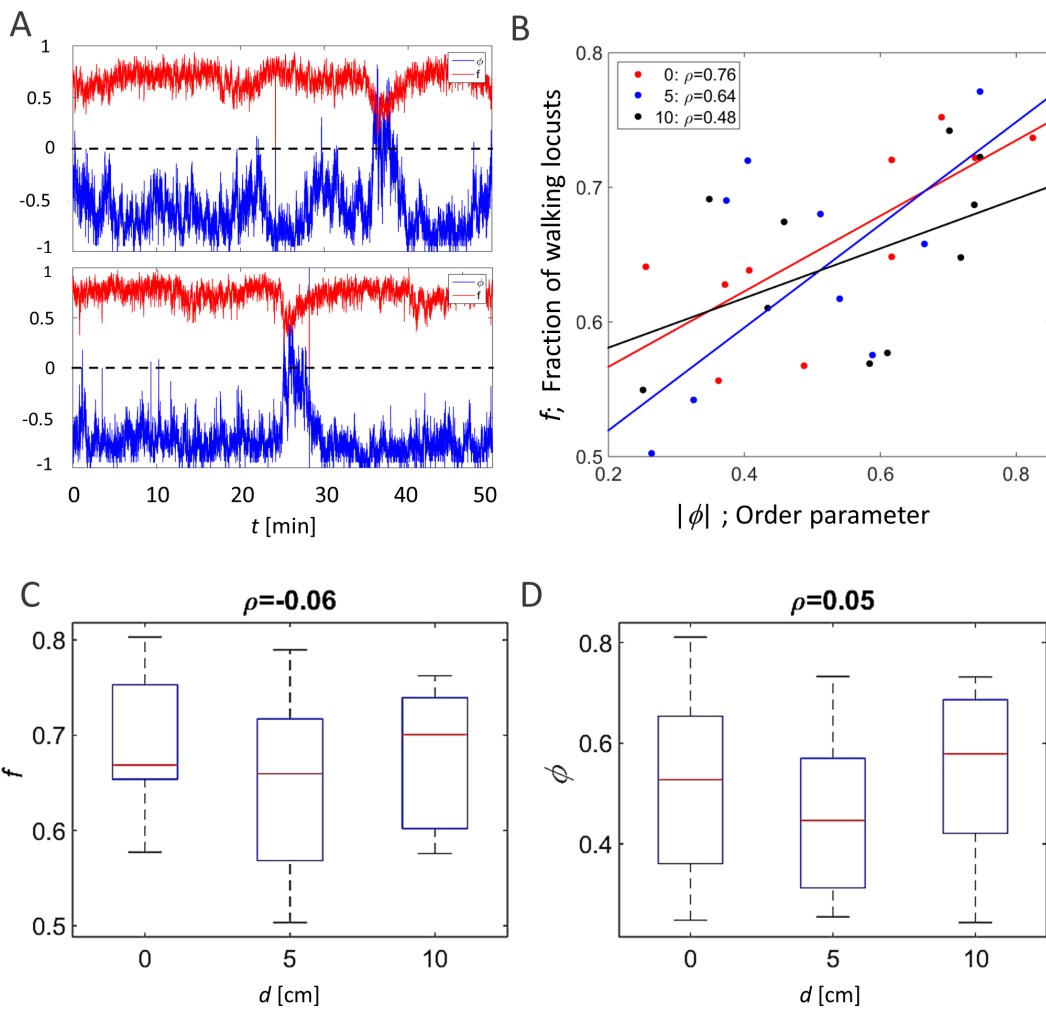

**Figure 2 Global parameters of the locust collective motion.** (A) The correlated changes in the order parameter, $\varphi$ (blue), and the overall fraction of walking locusts, $f$ (red), during a 50 min section of two experiments. (B) The correlation between the size of the order parameter, $|\phi|$, and the overall fraction of walking locusts, $f$, in all the experiments with the different arenas, as depicted in Fig. 1B: symmetrical (red), 5 cm (blue) and 10 cm displacement (black). $N = 10$ each. Linear regression is shown in solid lines and the significance of the regression is noted. (C) The overall fraction of walking locusts, $f$ (C1) and the calculated average order parameter, $|\phi|$ (C2) in the different arenas. No arena-type-related differences were found and the non-significant correlation is noted.

## Effect of changes in topography as reflected by comparing different parts of the arenas

In contrast to the above findings, a comparative investigation of the local environments within the different experimental arenas (the wide vs. narrow half or, alternatively, the narrowing vs. re-widening section; Fig. 1B) revealed clear effects of the changing topography. Each arena was divided into two equal-area halves. In the case of the symmetrical condition this resulted in two similar-interchangeable halves. In the asymmetrical-heterogeneous conditions, one section contained the funnel and the other section constituted the wider area (Fig. 1B; left and right halves of the arena, respectively).

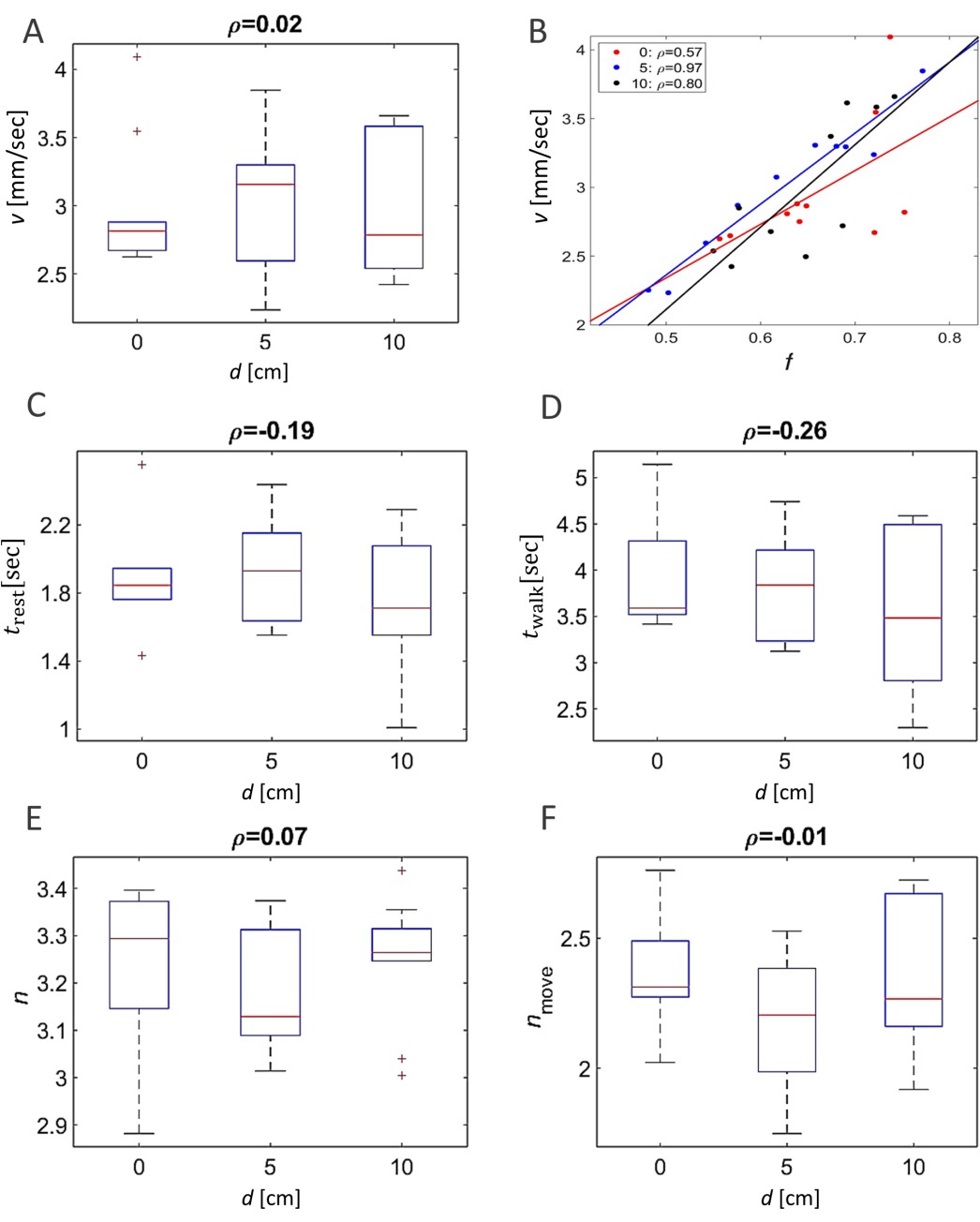

**Figure 3 The kinematics of marching were not affected by arena topography.** No arena-type-related differences were found in average walking speed (A). However, significant correlations ($p$-value $< 0.05$) were found between the locust average walking speed and the overall fraction of walking locusts (B) in all three arena types (marked with different colors). No arenatype-related differences were found in the average duration of resting (C) or walking (D) bouts. No arena-type-related differences were found in the average number of overall neighbors (E) or walking neighbors (F) within a short range around a focal locust. The correlation coefficients are noted above each graph.

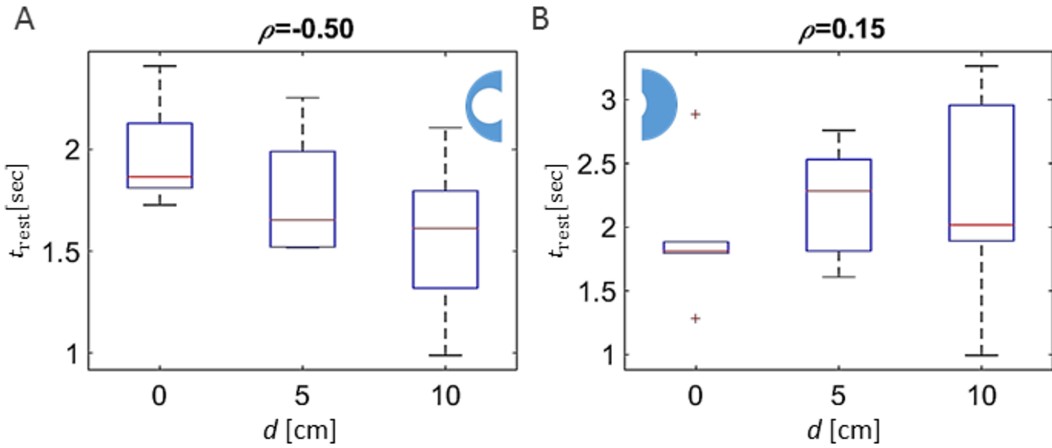

**Figure 4 Dependence of rest duration on arena topography.** (A) A clear arena-type-related difference in the average duration of resting (pausing) bouts was found when comparing the arena halves containing the funnel (in the symmetric, 0 displacement arena, one half was selected randomly). (B) A similar comparison of the opposite half of the arena yielded a non-significant correlation. The correlation coefficients are noted above each graph.

The duration of a pausing bout showed a clear dependency on the type of experimental arena when comparing the arena sections containing the funnel, i.e., the pause duration was shorter as the funnel became narrower (Fig. 4A, $p$-value 0.02). However, although less pronounced, the correlation was opposite in direction when comparing the opposite halves among the different arenas (i.e., longer pause durations in the wider parts of the asymmetric arenas; Fig. 4B). In contrast to the clear dependence of rest durations on the arena topography, analysis of the effect on walking durations gave inconclusive results ($p$-values 0.18 and 0.21 for the left and right halves). This is in accordance with our previous finding that locusts modulate their rest durations rather than their walking durations (reported previously for a homogeneous environment; *Ariel et al., 2014*).

Focusing on the heterogeneous-asymmetric arenas only, we further compared the kinematics of the locust marching, measured in the arena-half containing the funnel, to those in the arena-half characterized by the wider path (see Fig. 5A). The Most pronounced differences were those found in the arena resulting from the largest displacement (narrowest funnel; the right-hand topography illustrated in Fig. 1B) in respect to the locusts' average walking speed (Fig. 5B, $p$-value $< 10^{-8}$), and the distribution of the durations of the walking and pausing bouts (Figs. 5C and 5D; $p$-values $< 10^{-7}$). As can be seen, around the funnel the locusts' speed tended to be lower, with rest or stop durations shorter, but walking durations longer. Interestingly, only a few differences in kinematic parameters were found when comparing the section of the arena containing the narrowing part of the funnel with that containing the widening section (Fig. 5E). Only the mean speed of the locusts was found to differ (i.e., somewhat increased speed when emerging from the funnel, Fig. 5F), while the durations of a walking or pausing bout tended to be more similar (Figs. 5G and 5H ; $p$-values $> 0.05$).

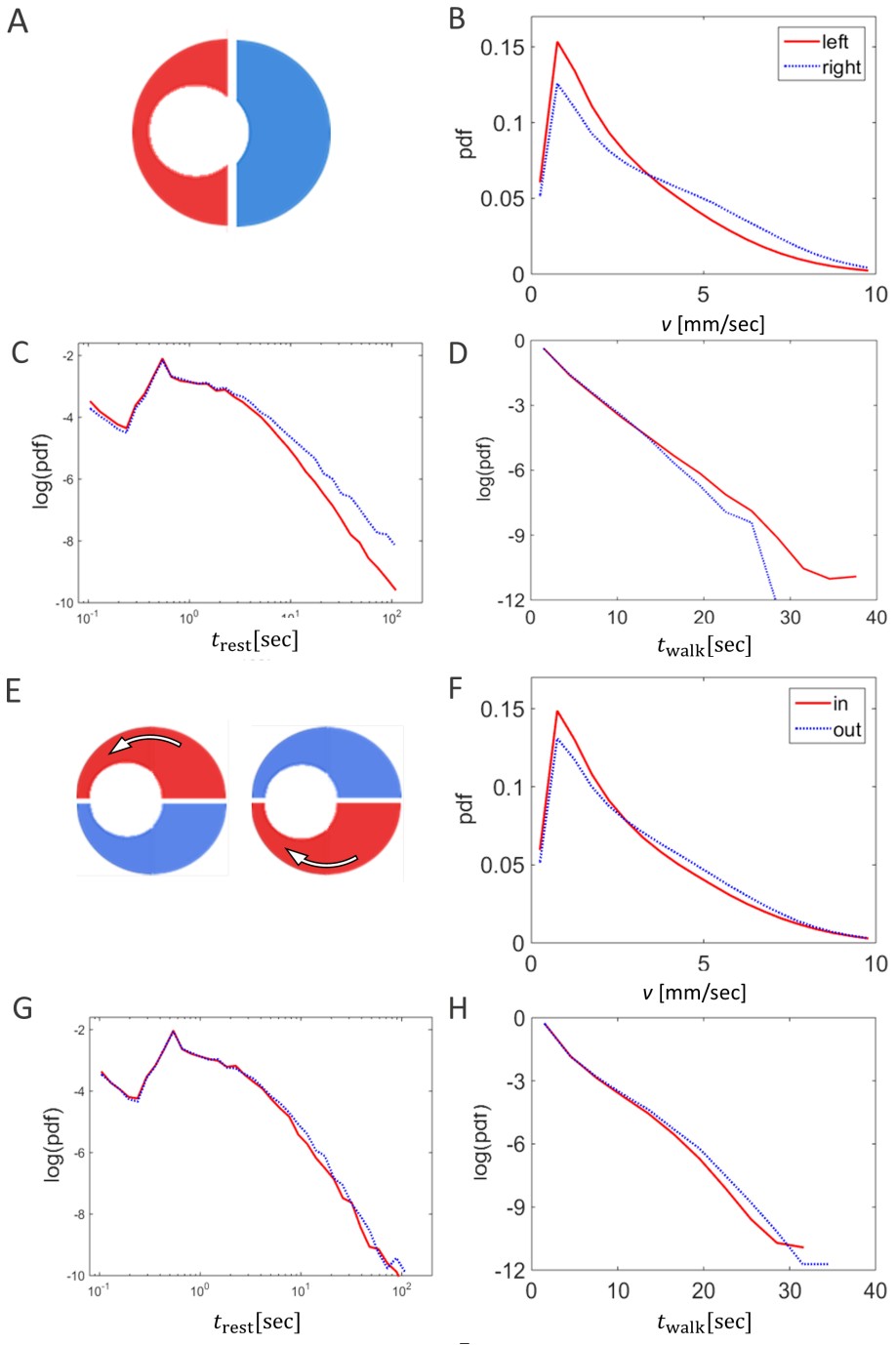

**Figure 5  Density heat maps and their analysis.** (A) Example heat maps for all three arena types. White regions denote frequently visited areas. (B) Schematic sketch showing a cut through heat map, translated into a graph. Density along the cut is rescaled to values between 0 (no locusts) and 1 (maximal observed density). The abscissa denote the location along the section, with the perimeter of the arena on the right of the graph and its center on the left of the graph. (C, D) Density through similar cuts as in (B), averaged over all experiments in each arena type. (C) Density of animals. (D) Density of moving animals only. Shaded areas are explained in the text.

The parameter most affected by the arena topography was found to be the local density of the locusts, as reflected in the number of locusts within a short range from a focal locust. For example, the distribution of close-neighbors count or that of close-moving-neighbors count (comprising only walking insects) was very different in the two halves of the arena formed by a 10 cm displacement (Figs. 6A and 6B $p$-value $< 10^{-8}$). This suggests that the marching locusts do not maintain constant distances from one another, but, rather, tend to temporarily crowd together in response to a narrowing environment. Furthermore, the increase in neighbors-count was found to be positively correlated with the displacement in the arena half containing the funnel, i.e., negatively with the width of the funnel (Figs. 6C and 6D, $p$-values $< 10^{-4}$). In contrast, in the opposite, wide half of the arena, the neighbors-count decreased with the displacement (Figs. 6E and 6F, $p$-value $< 10^{-4}$), suggesting a tendency of the locusts to spread out and fill the available space (see also Fig. 7, showing an example of locust trajectories in the re-widening section of the arena).

**Effect of changes in the topography as reflected in density heat-maps**

Figure 8A shows three examples (one for each of the three experimental conditions) of a heat map generated by attributing to each pixel in the arena a color related to the cumulative number of times it was occupied by a locust during the experiment—from black (unoccupied) to white (most frequently visited). Next, we converted the color code to a value between 0 (unoccupied) to 1 (most frequently visited ), and generated a plot of a slice of the mid-wide half of the arena (Fig. 8B). The resultant mean plots from the different experimental conditions were then overlaid to reveal similarities and topography-dependent differences (Figs. 8C and 8D). Three clear features of the locust motion in the different topographies could be observed:

1. Under all the experimental conditions most of the marching took place close to the perimeter of the arena (shaded dark in Figs. 8C and 8D). This is probably due to a combination of its circular nature and the reported tendency of locusts to minimize changes in their trajectory while actively marching (*Ariel et al., 2014*). The three plots (different experimental condition) are very similar in this respect.

2. Nonetheless, it is also clear that the locusts were using the entire space available to them (compare plots in the light shaded areas in Figs. 8C and 8D, and see also Fig. 7). This is mostly seen in the plot of the moving locusts (Fig. 8D).

3. Irrespective of the arena topography, or of the space available to the locusts, the level of marching near the inner wall of the arena was the same (compare the peaks within the light shaded areas in Figs. 8C and 8D).

## DISCUSSION

The principal findings of the current study were somewhat unexpected and even surprising. We refer to the limited extent to which the changing topography affected both, the macroscopic or system-level features of the marching locust group (order parameter and fraction of walking individuals), and the marching kinematics (average speed, duration of walking or standing bouts) and more (Figs. 2 and 3). Our experimental manipulation involved narrowing the path available to the locusts by a factor of 3 (from 15 cm in the

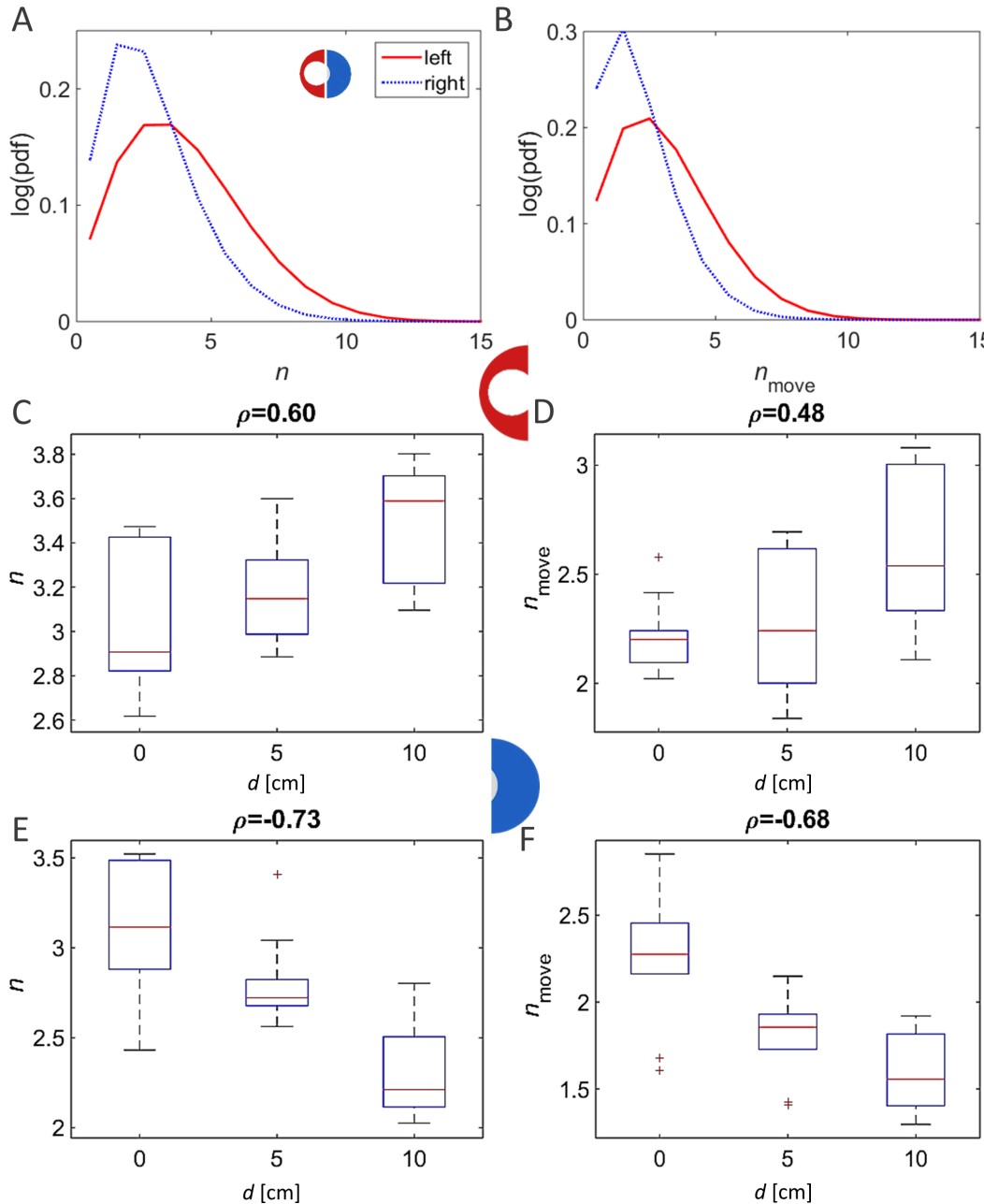

**Figure 6** **A comparison of locust densities in the narrow and wide halves of the arena.** As expected, the number of neighbors (both total as well as walking only) within 6 cm of a moving focal animal is highly dependent on the displacement (type of arena). (A) Distribution of the number of neighbors (left) or moving neighbors (right) in the arena with a 10 cm displacement (narrowest funnel; curve colors match the halves of the arena in the enclosed schematic drawing). (B) Average number of neighbors in each experiment (narrow half). (C) Average number of neighbors in each experiment (wide half). The correlation coefficients are noted above each graph.

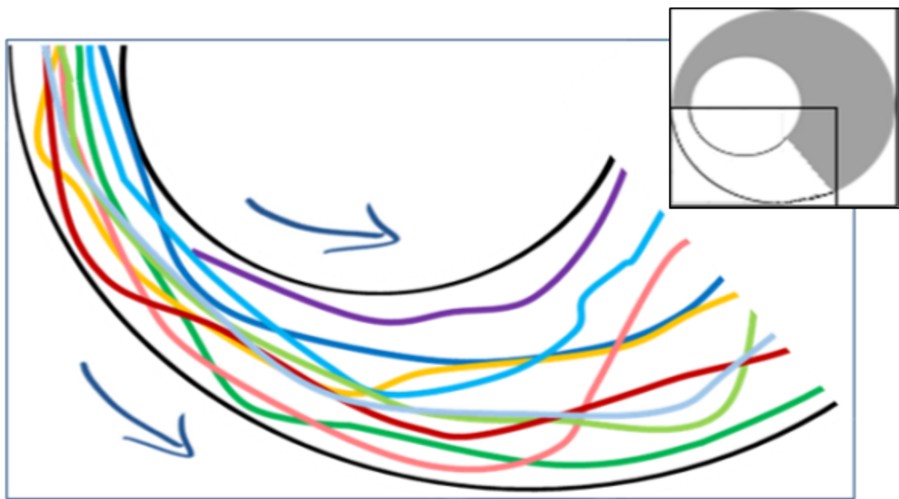

**Figure 7 Example trajectories of locusts leaving the funnel (overall 10 nymphs passing within a few seconds through the section marked white in the inset).** A tendency of locusts to spread out and fill the available space can be observed.

symmetrical arena to only 5 cm in the case of the 10 cm displacement). However, despite a clear increase in local densities, this funnel-like obstacle was insufficient to disrupt the order and collective movement, at least within the tested experimental parameters. These findings serve as a strong indication of the stability and robustness of locust collective behavior and its tendency to adapt in order to maintain the marching activity.

It was only when comparing between local sub-environments (the two halves of the experimental arena) that the effects of the funnel-like topography, or the opposite wider space, became evident (Figs. 4–6). A swarm of marching locusts encountering a funnel like structure in their environment could respond in several possible manners. These include changes in the behavior of the individual locusts as well as in the swarm dynamics. The individual locust may slow down or, alternatively, speed up when encountering an obstacle (a wall or a standing conspecific in our case) in order to bypass it. The locusts may increase their density, i.e., reduce the average distance between conspecifics (Fig. 9A), or change the relative position towards one another (aligning or forming queues), and more. Similarly, at the level of the swarm, various responses could be envisioned for a marching swarm exiting a funnel or experiencing a widening of its path. In this case the two major alternative behaviors would be to spread out and fill the available space (Fig. 9B), or to show a kind of memory of their path history as reflected in the shape of the swarm, i.e., to refrain from spreading out when exiting the funnel (Fig. 9C).

Our study provided two important insights. The first: that the locusts tended to spread out and fill the space available to them (as in Fig. 9B rather than in Fig. 9C). Hence, there should be a theoretical and practical scenario in which this behavior will be opposed or limited by the strong and instrumental tendency of gregarious locusts to aggregate (this limit was not reached in the current work, but may appear at higher animal densities). As we have shown, the funnel-like topography and the narrowing of the path resulted in alteration of the marching kinematics, such that both speed and pausing durations were

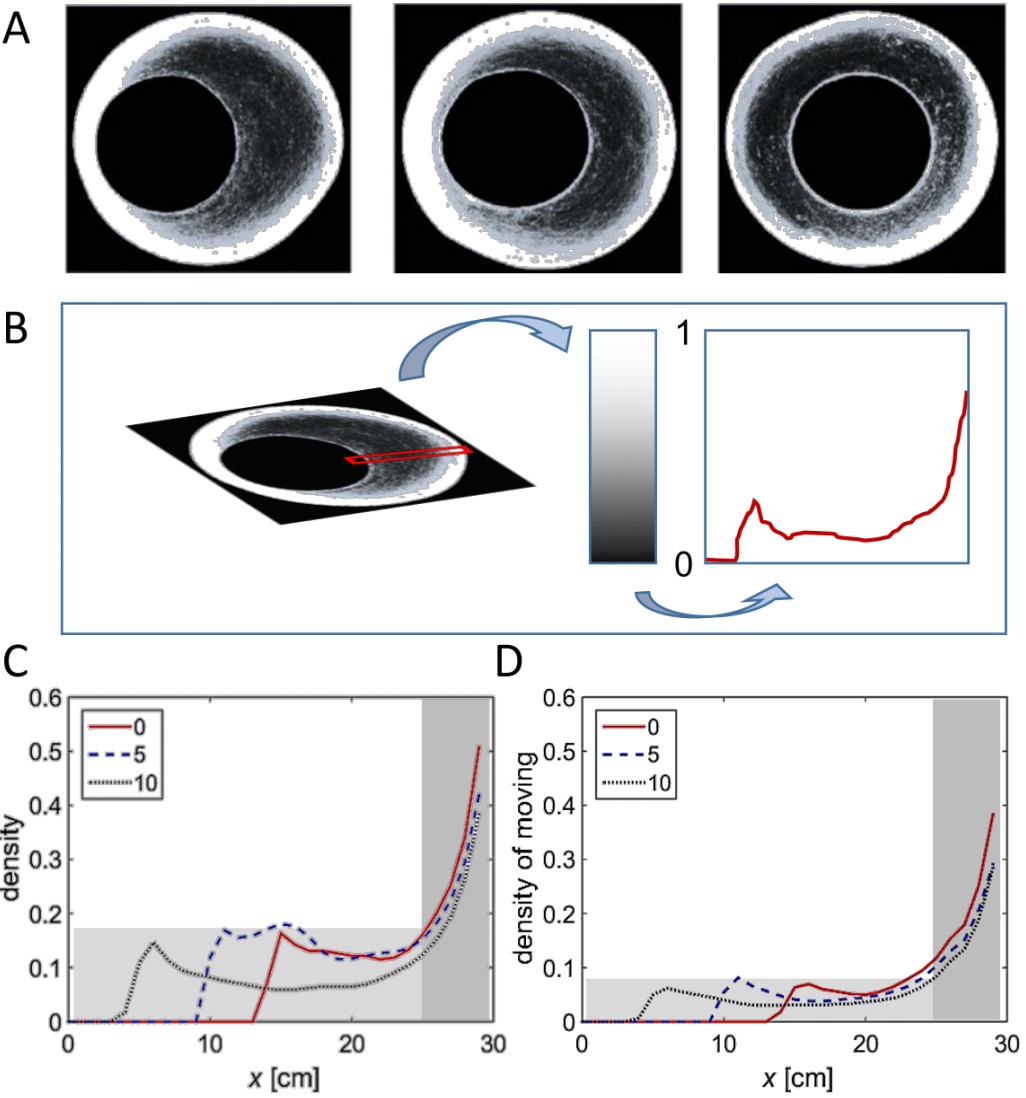

**Figure 8  Density heat maps and their analysis.** (A) Example heat maps for all three arena types. White regions denote frequently visited areas. (B) Schematic sketch showing a cut through heat map, translated into a graph. Density along the cut is rescaled to values between 0 (no locusts) and 1 (maximal observed density). The abscissa denote the location along the section, with the perimeter of the arena on the right and its center on the left. (C) Density through similar cuts as in (B), averaged over all experiments in each arena type. 1. Density of animals. 2. Density of moving animals only. Shaded areas are explained in the text.

reduced. Our previous findings, suggesting that it is the locusts' rest durations rather than walking durations that are typically modulated in order to adjust the swarm dynamics (*Ariel et al., 2014*), are thus confirmed, supporting the instrumental role of the intermittent motion (pausing) in the marching dynamics.

The second prominent feature demonstrated by the locusts in our experimental arenas was that of their attraction to the inner wall. What might appear to be attraction to the outer wall could be, at least partly, a result of a tendency to maintain the walking trajectory (*Ariel et al., 2014*). Many animals, insects included, exhibit an attraction to walls while moving

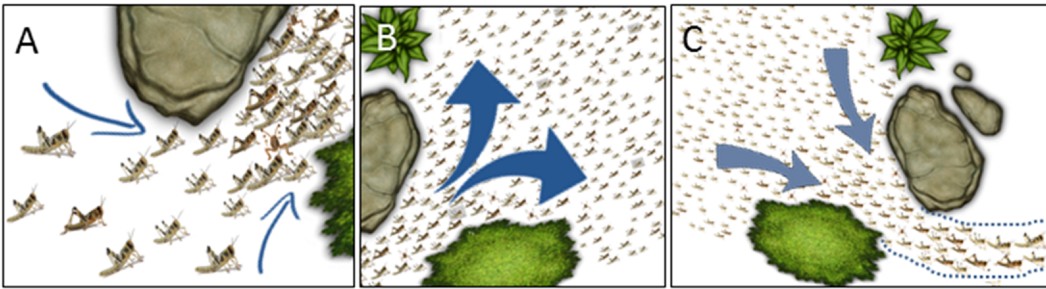

**Figure 9** **Cartoon showing possible swarm behavior in nature.** (A) Converging into a funnel results in increased density. (B) Spreading out of a funnel. (C) Exiting a funnel while preserving the swarm's shape. See further details in the text ('Discussion').

in a defined space (see for example cockroaches: (*Jeanson et al., 2005*), or flies: *Soibam et al., 2012*). The tendency to orient in space by means of mechanical contact is termed "thigmotaxis" (*Fraenkel & Gunn, 1962*). Thigmotactic animals have a higher probability of walking adjacent to walls, although different curvatures of walls may have different effects: concave wall has a higher "attraction level" than a convex one (*Creed & Miller, 1990*). A tendency to follow walls (or, alternatively, a crest or ridge in the landscape) could potentially be advantageous for swarm maintenance, by way of increasing the likelihood of encountering a conspecific, and could thus be important in the collective behavior (*Jeanson et al., 2003*).

It is important to note that there are probably species-specific differences related to the above features, i.e., the density within the swarm and, mostly, the shape of the swarm (e.g., *Uvarov, 1977*; *Hunter, McCulloch & Spurgin, 2008*; *Buhl et al., 2011*; *Ariel & Ayali, 2015*). Species may also differ in the landscape- or topography-dependent effects on the behavior of individuals and on the consequent changes in the spatial attributes of the swarm. In accordance with our present findings, *S. gregaria* marching bands under natural conditions have indeed been reported to spread out widely along very large distances, in contrast to the long ribbon-like structures reported for other species (*Kennedy, 1939*; *Kennedy, 1951*; *Uvarov, 1977*).

One should of course also be cautious when extrapolating from the results of laboratory experiments to the natural behavior of locusts in the field. The current study constitutes a first step in exploring the important yet very broad question of the effect of topology on the collective dynamics of marching locusts. Hence, we chose here to concentrate on a relatively straightforward regime of experimental conditions, i.e., a specific number of animals and an arena area that had been found to be favorable for promoting locust marching. In other words, the experimental conditions were chosen so that the only disrupting factor for the marching animals would be the narrowing funnel. The effects of other sources of disruption to the coherent marching (for example, over- or under-crowding, large or small distances from the walls, different nutrition etc.), may be non-additive and could significantly change some of the conclusions of the current study. Nonetheless, our results show that even under "optimal" marching conditions, a simple topological heterogeneity has an interesting and somewhat unexpected impact on the locust dynamics.

Finally, we emphasize our findings regarding the capacity of locusts to adapt their response to their surroundings (the swarm dynamics) in order to compensate for barriers and bottlenecks in their path. Such behavioral strategies can explain the non-intuitive independence of the order parameter from the funnel width. This experimental finding contradicts several previous theoretical predictions, which suggested that environmental heterogeneities can qualitatively alter the dynamics of collectively moving particles in a complex way—decreasing the flux through the bottleneck, promoting particular swarming patterns, and even changing the phase-behavior of the system (see Introduction for details). Our results show that, at least under the density regimes tested, persistent and stable dynamics can be promoted by means of individual adaptability. For example, whereas *Shklarsh et al. (2011)* demonstrated that individual adaptability can enhance search efficiency in swarm optimization methods, our results suggest that adaptability can lead to an opposite effect and increase robustness and resilience. Future experimental and theoretical studies should thus investigate organisms and models with performance-adaptable interactions.

### Funding

This work was supported by an Israel Ministry of Agriculture grant (891-0277-13) to Prof. Amir Ayali and an NSF Research Network on kinetic equations (KI-Net) (1107444 and 1107465) to Dr. Gil Ariel. The funders had no role in study design, data collection and analysis, decision to publish, or preparation of the manuscript.

### Grant Disclosures

The following grant information was disclosed by the authors:
Israel Ministry of Agriculture: 891-0277-13.
NSF Research Network on kinetic equations (KI-Net): 1107444, 1107465.

### Competing Interests

The authors declare there are no competing interests.

### Author Contributions

- Guy Amichay performed the experiments, wrote the paper, prepared figures and/or tables, reviewed drafts of the paper.
- Gil Ariel conceived and designed the experiments, analyzed the data, contributed reagents/materials/analysis tools, wrote the paper, prepared figures and/or tables, reviewed drafts of the paper.
- Amir Ayali conceived and designed the experiments, wrote the paper, prepared figures and/or tables, reviewed drafts of the paper.

### Data Availability

Ayali, Amir (2016): Asymmetric arena.MTS. figshare.

10.6084/m9.figshare.3545844.v1;

Ayali, Amir (2016): Symmetric arena.MTS. figshare.
10.6084/m9.figshare.3545841.v3.

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
