# Peer review of "The effect of changing topography on the coordinated marching of locust nymphs"

_PeerJ, doi:10.7717/peerj.2742_

## Round 0.1 · original submission · Major Revisions

· Academic Editor

Major Revisions

In addition to the Reviewers' comments, I note the following:
28: Change ‘by which’ to ‘to which’.

197: The value of the correlation coefficient gives only the strength of the relationship between two variables, not the significance. Therefore, it is incorrect to assume significance if the correlation coefficient is greater than 0.3.

217, Figure 3A: The group at 0cm seems to have a different distribution than the other two groups. It would be worth testing these data for possible differences in distribution.

348: Change ‘into organisms and models’ to ‘in organisms and models’.

·

Basic reporting

The group of Amir Ayali focuses on the mechanism of collective motion, and has published a series of papers about this topic. Here this study reported that the change of topography does not change the synchrony of the collection motion of S. gregaria. Thus, this study provides new insight into mechanism of collective behavior, in contrast to other reports about the theoretical results predicting that environmental heterogeneities qualitatively alter the dynamics of collectively moving particles.
This manuscript has been done well on the whole, but I indeed have some points for this work.
1) In the abstract, this sentence “The effects of these traits on the subsequent dynamics of the locust swarm are discussed” in the line 37 in abstract should be deleted. The last sentences provided the content section of this manuscript.
2) In the introduction part, the paragraph 3 (line66-77) describes the information about this study, and does not conform to the logic flow between paragraph 2 and paragraph 4.
3) In Fig.9B, after locusts pass by tunnels, they will enter open areas and spread out. This behavioral pattern is very different from those patterns observed in the limited cycle arena field in the lab. The differences should be discussed and pointed out in discussion section.
4) In ling 153, “where” would be “were”;
5) Line 263, “while” would be “white”;
6) So many mistakes in punctuation mark in references.

Experimental design

In methods, why was the image of the arena divided into 248*248 pixels? The authors would provide the rationality.

Validity of the findings

In statistics, there is lack of the probability value (P value) of correlation for significance in calculating ρ value. “Correlations were considered as significant for | ρ | > 0.3” in method section is somewhat arbitrary.

It would be helpful if authors perform two statistical comparisons, 5cm vs. 0 cm, 10cm vs.0 cm, to independently detect the effects of 5 cm displacement and 10 cm displacement in all experiments, and finally provide the statistical P values.

Reviewer 2 ·

Basic reporting

In general, the scientific language of the paper is poor. The English is not clear and presents some basic grammar errors in the structure of the sentence especially in the Results and Discussion sections (for example see the first two sentences of the Discussion). I encourage the authors to re-write the paper and to ask a native speaker to check the English.
The Introduction includes sufficient background and prior literature is appropriately referenced. Only one sentence (lines 117-120 page 5) is very confusing and should be rephrased as well as expanded since it is not clear what the "key questions" are and how they "may be different in details" (Which details? May they be different in details depending on the species?). Please clarify.
The thing that puzzles me the most is that the body text doesn't contain any statistic. The results are presented as significant or not without reporting any p value in the text; only the rho values of the correlations are reported on top of the plots in the figures.
The Figures captions are not very well described and can often confuse the reader because of the way the plots have been labelled. The authors should be consistent with that (A1, A2, B1, B2 and so on) through all the figures. Apart from the values of rho, the p values should be reported as well.

Experimental design

The research question is quite clear and fits well in the scientific background described in the introduction. The experimental design is adequate to answer the research question but it is not very clearly presented in the "Behavioural setup" paragraph. The authors should be consistent with the terminology (condition, group, set-up). It should be stated more clearly how many conditions they tested and in how many different groups of animals, i.e. 5 different groups of animals for each experimental condition, where for experimental conditions the authors mean set-up conditions. Maybe numbers or labels for these different conditions may help. In particular, the last sentence of the "Behavioural setup" paragraph is very confusing.
Moreover, it is not clear why the authors selected only the last 200,000 frames of each recording (line 153 page 7). The reason should be mentioned and justified. Similarly, it should be explained why they chose to focus on a specific range of distances (i.e. 6 cm from each focal animal - line 188 page 8).
"To normalize the differences between the experiments" (lines 199-200 page 9): please clarify if this refers to the 5 different groups of locusts tested in each experimental condition.

Validity of the findings

The data seem robust and controlled, but not statistically sound. It is not clear why the authors analysed the data by looking at correlation coefficients instead of simply comparing the means of the different parameters in the different conditions with T-tests. The results of the statistical analyses should be reported in the text as well as on the figures. The results are not well presented due to grammar issues mainly. It should be highlighted better what are the findings by writing what was found to be significantly different.
"This is in accordance with... (line 235-237, page 10): in which experimental conditions was this previously observed? Please explain.
Heat-maps: it should be mentioned what white and black correspond to in the grey-scale (i.e. black = maximum occupied; white = unoccupied). The whole section of the "heat maps" is pretty confusing. Highlighting points (1,2,3) should help understanding the findings, but in this case the three points don't follow each other in a clear way and they are within sentences instead of being at the beginning of each sentence.
In the Discussion, please explain better what "show a kind of memory of their path history as reflected in the shape of the swarm" (lines 299-300, page 13) refers to.
"As noted" (line 324, page 14): please add the appropriate reference.
The last sentence of the Discussion about future directions is not clear and needs to be explain better.
In general, as mentioned already, the grammar of the sentences should be checked and several sentences need to be rephrased.

---

## Round 0.2 · Minor Revisions

· Academic Editor

Minor Revisions

Please address the remaining minor revisions listed by Reviewer 1.

·

Basic reporting

The quality of this manuscript has been greatly improved.

Experimental design

The description of ANOVA analysis is not provided in this manuscript, such as in the method section or figure legends. The authors may check the description of this statistical method.

Validity of the findings

The p-Values are provided in the results, but it is confusing that the p-Values are calculated by ANOVA or correlation methods. For example, in line 220, ‘p-Values 0.57 and 0.53’, is p-Values 0.57 for fig 2C1, is p-Values 0.57 for for fig 2C2? Do these p-Values indicate for correlation or the comparisons among three groups (0, 5, 10 cm)? The style of p-Values and their description would be consistent and clear in the manuscript.

In line 241, are three p-Values for fig4?

Line 265, ‘p-Value <0.016’ is very confusing. If the authors intend to provide specific p-values, the style would be ‘p-Value = 0.016’. If to provide the range of p-values, the style would be ‘p-Value < 0.05’. The authors may check the description of the styles in the whole manuscript.

Reviewer 2 ·

Basic reporting

The authors replied to all my comments and concerns in an appropriate way. I can now recommend the manuscript for publication.

Experimental design

The authors improved this part as well as the readability of the manuscript in general.

Validity of the findings

The authors have been very receptive to my comments and considered them
carefully. I am particularly impressed by the implementation of the
statistical analysis I suggested.

---

## Round 0.3 · accepted · Accept

· Academic Editor

Accept

Thank you for submitting an interesting paper.